# Turning the spotlight: Hostile behavior in creative higher education and links to mental health in marginalized groups

**Marina Fischer**[1,2]*, **Susanne Veit**[2,3], **Pichit Buspavanich**[1,4,5], **Gertraud Stadler**[1,6]

**1** Research Unit Gender in Medicine, Charité–Universitätsmedizin Berlin, Berlin, Germany, **2** Research Unit Good Work in a Transformative World, Berlin Social Science Center (WZB), Berlin, Germany, **3** Cluster Data-Methods-Monitoring, German Center for Integration and Migration Research (DeZIM), Berlin, Germany, **4** Institute of Sexology and Sexual Medicine, Department of Psychiatry and Neurosciences, Charité–Universitätsmedizin Berlin, Berlin, Germany, **5** Department of Psychiatry, Psychotherapy and Psychosomatics, Faculty of Health Sciences Brandenburg, Brandenburg Medical School Theodor Fontane, Neuruppin, Brandenburg, Germany, **6** Institute of Applied Health Sciences, University of Aberdeen, Aberdeenshire, United Kingdom

* marina.fischer@charite.de

**Data Availability Statement:** The provision of all data is not possible due to its sensitivity (sensitive human research participant data, large number of socio-demographic data that allow re-identification

## Abstract

Hostile, discriminatory, and violent behavior within the creative industries has attracted considerable public interest and existing inequalities have been discussed broadly. However, few empirical studies have examined experiences of hostile behavior in creative higher education and associated mental health outcomes of early career artists. To address this gap, we conducted a survey among individuals studying at higher education institutions for art and music (N = 611). In our analyses of different types of hostile behaviors and their associations with mental health and professional thriving, we focused on differences and similarities between marginalized and more privileged groups across multiple diversity domains. A substantial percentage of participants reported hostile behaviors in their creative academic environments. Individuals from marginalized groups reported more hostile behaviors, which partially explained their worse mental health and lower professional thriving. These findings indicate a clear need for the creative sector to implement strategies to create safer environments, particularly for early career artists from specific socio-demographic backgrounds. We conclude by suggesting strategies for prevention in this highly competitive industry.

## Introduction

The 2017 #MeToo movement exposed a broad range of sexual misconduct and power-abusive incidents across social backgrounds and industries. The initiative of the Black feminist scholar and activist Tarana Burke was brought to a wide public audience by prominent Hollywood personalities [1]. Previously implicit issues of hostile sexualized behavior in various areas of life found their way into public discourse through individual revelations, and in the months that followed, these reports were greeted by both, solidarity and backlash. A spotlight was

when crossed, vulnerable groups). Providing the whole data set is also not possible due to the regulations of the local ethics committee (https://ethikkommission.charite.de/en/) in accordance with the EU-GDPR and German data protection laws. We to our best knowledge assured to clarify that these ethical and legal restrictions apply in the open practices and data accessibility statement in the paper and in the accompanying data protection protocol upon submission. The data slice and all supplementing materials (survey material, code) are available on OSF: https://osf.io/stced. Interested fellow researchers can contact our ethics committee at any time for data-related information. We will do our utmost to provide researchers with an insight into our data. Here are the contact details of the Ethics Committee: Ethics Committee of the Charité - Universitätsmedizin Berlin Postal address: Berlin State Office for Health and Social Affairs Office of the Ethics Committee of the State of Berlin Turmstr. 21 House A 10559 Berlin Email: ethikkommission@charite.de Phone:+49 30 450 517 222.

**Funding:** MF received funding from the Hans Boeckler foundation (https://www.boeckler.de/de/index.htm) for this project (grant no. 411926). The funder did not play any role in the study design, data collection and analysis, the decision to publish, or the preparation of the manuscript.

**Competing interests:** The authors have declared that no competing interests exist.

shone on the underpinnings of sexualized violence and hostile behaviors particularly within the creative industries, and there were calls for fundamental societal changes.

## Hostile behaviors: Concepts and vulnerabilities

Sexual, psychological and physical violence are globally widespread forms of hostility towards women in particular [2]. In Europe, sexual harassment and aggression are prevalent among younger persons and more frequently affect marginalized groups [3, 4]. Researchers and policy institutions have noted the importance of having a broad conceptual perspective on violent behavior, its contexts and its consequences for individuals and organizations [5, 6]. Following sociologist Liz Kelly [7], violence is framed as a continuum beginning at much earlier stages than physical assault or rape, for instance with (gendered) discrimination [8], recurrent and subtly degrading micro aggressions [9], technologically facilitated violence [10], and sexual aggressions in forms such as harassment, coercion, and physical assault [11].

Social psychologists and social science researchers have proposed multi-level explanations for the phenomenon of sexualized violence. The socio-cultural approach suggests that macro-level power dynamics within societies enable and sometimes endorse sexualized violence against marginalized and/or underrepresented groups, such as marginalized based on gender [12]. At the meso level, researchers have examined structural aspects of institutions, including cultures and climates, as possible contributors to violence, such as in workplace and academic contexts [13]. In the case of higher education institutions (HEIs), the literature indicates significant rates of victimization and a lack of effective prevention measures [14, 15]. Micro-level analyses of violent and hostile behaviors in certain environments additionally identify situational aspects and individual characteristics of perpetrators, affected persons, and bystanders [12]. These characteristics, in turn, can appear in various combinations that require intersectional analyses.

Intersectionality-sensitive perspectives consider individual positions of persons on various intertwined axes of power, privilege, and marginalization within societal and institutional structures that come into play in violent and abusive situations. This approach offers a fine-grained view of group-related vulnerabilities [16, 17] regarding individual positionalities and experiences connected to intersecting aspects of marginalization. Belonging to certain marginalized groups, such as persons identifying as trans, inter, or non-binary, is linked to poorer health outcomes [18]. The minority stress model, initially developed for the LGBT community, suggests that marginalized groups experience an additional burden due to discriminatory social dynamics and stigmatization [19] and lack social safety in their daily lives [20]. Women, LGBTIQ+-individuals, migrants and refugees, persons with mental health conditions, and younger people are at a higher risk of experiencing sexual violence and harassment. Multi-marginalized individuals are especially affected [4, 8].

### Creative working environments

Sexual violence, abuse of power and discrimination can lead to poorer mental and physical health among workers [21]. In its Violence and Harassment Convention [22], the International Labour Organization commits the ratifying members to "respect, promote and realize the right of everyone to a world of work free from violence and harassment" (Article 4). As violent behaviors are widespread, they are particularly common in some environments, e.g. health and social care, politics, education, and the cultural and creative industries [13].

The creative industries enable power-abusive behaviors due to certain characteristics, including the importance of informal networks, strong competition for highly insecure jobs, high time pressure, imbalanced (gendered, racialized, classed, ability-based) power relations

and steep hierarchies [23, 24]. Dominant narratives often justify insecure, precarious work under unwelcoming, harmful conditions with passionate motivation and non-monetary rewards through creative output and social capital [25]. This particularly affects the already marginalized and precariously employed individuals in this industry and the and disproportionate demands placed on them, for example because of their gender identities [26, 27]. In recent years, a growing body of literature has emerged addressing issues of inequality and abuse of power in various institutional cultures and fields of creative labor [28–33].

## Misconduct and violence at creative higher education institutions

The #MeToo movement and the public debates surrounding misconduct cases at art and music colleges have also prompted long-overdue transformation processes at these institutions [34]. At European creative HEIs (note that in this article, we adopt the terminology devised by Comunian and colleagues [35], who referred to the concept of "creative higher education") this has led, among other reactions, to the increased establishment or redesign of complaint policies, equality, diversity and inclusion initiatives, and codes of conduct [34, 36, 37]. However, empirical research providing insights into specific experiences within these institutions is limited [37].

There are pioneering projects that shed light on vulnerable groups and harmful behaviors in creative HEIs [38], some of which specifically address unbalanced power dynamics and institutional cultures [39, 40]. Despite the transformational claims of the overarching creative industry, creative higher education environments partly fail to provide a truly inclusive and welcoming space for people from marginalized groups [41, 42]. HEIs play a crucial role in socializing future creative workers into professional norms on proximity and boundaries [43]. Beyond formal curricula, there are "invisible practices" [39] that implicitly produce and provide knowledge regarding authoritative dynamics and roles within the existing hierarchies of both the institution itself and the surrounding industry. It is essential to gain a more comprehensive understanding of early career artists' experiences to enable specifically tailored policies and prevention measures that effectively address the needs of vulnerable positions in these complex institutional and professional cultures. Understanding the early career situation of marginalized creatives is especially relevant due to the excess minority stress burden and its impacts on mental health, as these individuals may be particularly affected by harmful practices.

To the best of our knowledge, no study to date has linked experiences of violence at German creative higher education institutions with aspects of students' mental health and artistic development opportunities. In particular, we are not aware of any such study that has looked at it from the perspective of individuals facing (multiple) marginalization. Our study aims to fill this gap by applying a diversity-sensitive perspective to experiences and consequences of hostile behaviors in creative higher education environments, and exploring specific needs for future intervention programs in early artistic careers based on the findings. Drawing on a cross-sectional online survey, we addressed several hypotheses (see Fig 1; detailed hypotheses: https://osf.io/stced).

H1: Among early career artists, diversity domain members (i.e., non-cis male persons, non-heterosexual persons, younger people, people with a family history of migration and/or identifying with an ethnic-racial minority, people with disabilities, mental health issues and/or physical health issues) report higher rates of hostile behavior experiences.

H2: Hostile behavior experiences are negatively linked with mental health and well-being.

H3: The link between hostile behavior experiences and mental health and well-being is moderated by diversity domain membership.

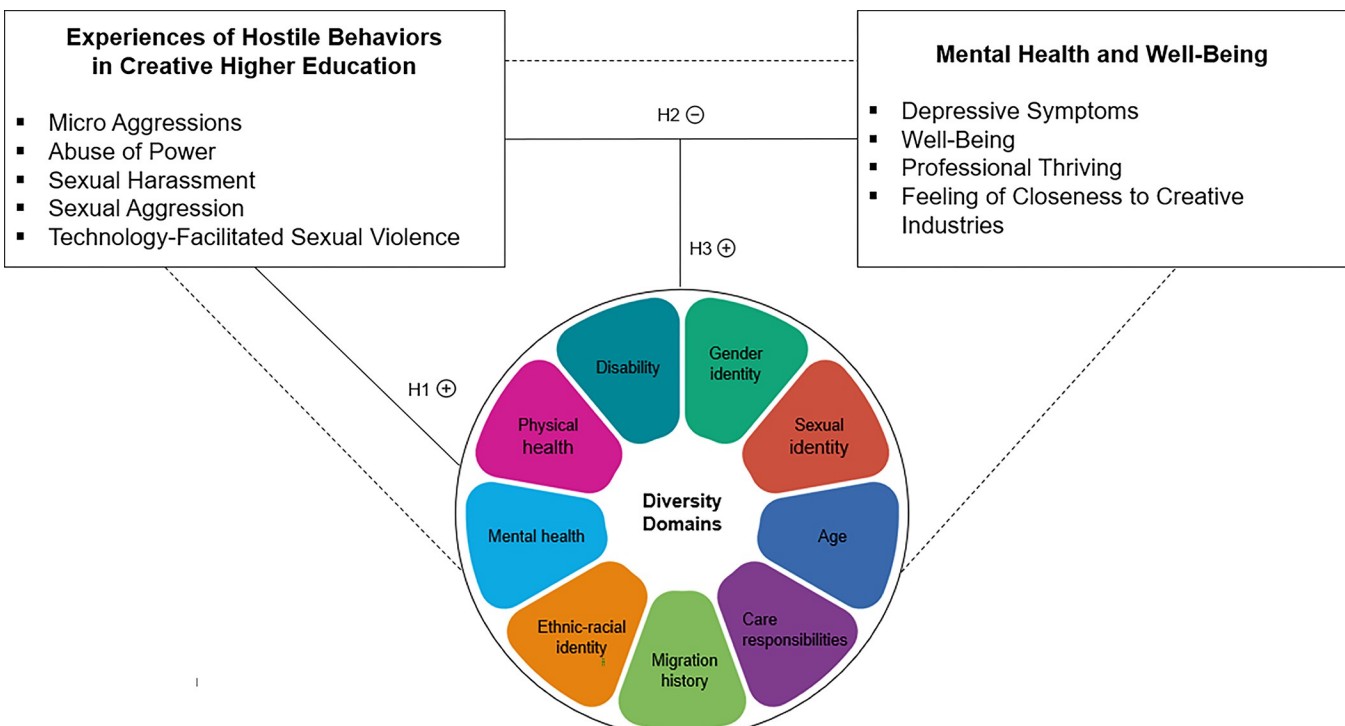

**Fig 1. Hypothesized associations between experiences of hostile behaviors in creative higher education, mental health and well-being, and diversity domains.** Solid lines: assumed correlations (H1, H2) and moderation (H3), dashed line: mediation (examined exploratively). Plus and minus signs: positive and negative links.

In addition to the preregistered hypotheses above, we exploratively analyzed the mediating roles of hostile behavior experiences on mental health and well-being in creative professionalization, depending on early career artists' diversity domain identification.

## Methods

### Researchers' positioning

Our author team consists of three psychologists and one psychiatrist. With backgrounds in health sciences, social psychology, social sciences, as well as counseling and medical practice with vulnerable groups, we bring a variety of perspectives to the issues raised in this research project. We believe this to be a strength of our contribution. Our disciplinary backgrounds and traditions have certainly informed and influenced our use of methods, analysis, and the theoretical framework of this study.

Against the backdrop of our research, all the authors are familiar with issues of organizational diversity, the lack thereof, and addressing structural issues related to these aspects. We also share a research interest in violent and discriminatory behaviors toward marginalized groups, and the ways they relate to aspects of mental health and prevention.

On an individual level, we bring different lived experiences of certain characteristics of diversity to this research project, including having a family history of migration, or being first generation academics. These experiences have been part and parcel of our personal processes of finding a place in hegemonic patriarchal contexts in general and higher education institutions in particular. Our perspectives are also shaped by cis identities. In addition to personal positionalities, our research perspectives are informed by experiences of patient and

stakeholder engagement practice with various marginalized groups. We see our work as strongly committed to understanding societal inequalities as intersectional and overcoming them.

## Open practices and data accessibility

This study has been pre-registered on the Open Science Framework (OSF), with study materials, questionnaire, analysis code and codebook publicly available via https://osf.io/stced. Primary data are not publicly available due to their sensitivity. To comply with transparency principles, we provide a data slice containing relevant variables that have undergone several thorough transformations and anonymization measures, and are thus de-identified. The slice is also available on the OSF repository.

## Procedure

Using a cross-sectional convenience sample design, we collected data online between July 18 and September 1, 2022. We sent out invitations to participate by e-mail to all 57 state-recognized higher education institutions of art and music education in Germany and approached equality officers and student representatives at each institution as multipliers. This resulted in a total of 148 personalized e-mail invitations. We also approached professional networks and initiatives of young creatives in Germany. To welcome participation and compensate for the length of the survey (approximately 20 minutes), we offered to enter a prize draw for 12 vouchers (total: €300) for online shops of participants' choice after completing the survey. Participants were required to provide active written informed consent to participate before entering the questionnaire. In line with recommendations for responsible research on sexual violence [43], we included a further disclaimer about potentially burdensome content and the possibility of ending the questionnaire at any given time without negative consequences. A list of partly sector-specific counseling services on the topics of violence and abuse of power was provided on the first page and listed on the footer of each questionnaire page. The Ethics Committee of Charité–Universitätsmedizin Berlin approved this study project (approval number EA2/295/21).

## Participants

In order to provide the most comprehensive representation of the artistic student body in Germany, we invited individuals from all degree programs to participate. On the one hand, this approach was necessary to safeguard the identity of students from niche subjects with occasionally very small classes. At the same time, we were able to gain insight into the overarching structures at German universities of the arts as specific ecosystems of early artistic careers. We aimed for a minimum sample of $N = 300$ participants. Due to the great willingness to participate during the announced study period (data collection was stopped after six weeks), we eventually reached a total of $N = 682$ persons. We excluded participants who indicated that they were not currently enrolled in a public art and/or music school in Germany ($n = 53$) and 18 participants due to missing information on their current subject(s) of study.

## Measures

For the online survey we partially adapted mostly existing and widely used instruments to the purpose and target audience of the study. For each item, we provided a "prefer not to answer" option.

**Diversity domains.** To address potential marginalization of participants, we used the Diversity Minimal Item Set [18], a concise instrument drawing on the concept of diversity and discrimination domains. Participants were asked to provide information on their gender identity, sexual identity, age, caregiving responsibilities, migration history (own as well as parents'), self-identification with ethnic-racial minority identities, psychiatric diagnosis, disability and (chronic) health conditions. Individual description fields were provided. Group membership was coded as 0 (belonging to respective marginalized group, e.g. lesbian, gay, bisexual, asexual, pansexual or describing individual sexual identification) or 1 (belonging to the respective privileged group, e.g. heterosexual). The Diversity Minimal Item Set also includes an item on discriminatory experiences within the past 24 months, which we adapted to address discrimination in creative higher education contexts.

**Micro aggressions.** We used seven items from micro aggressions scales targeting racialized ethnic minorities and sexual and gender minorities [44–46]. Participants were asked to rate whether they had experiences of specific everyday micro aggressive behaviors within their study context based on their identity or being part of certain groups on a 5-point scale (does not apply at all–applies completely), for example: "Because of my group affiliation or identity. . . my opinion or attitude was not addressed." For statistical analyses, we calculated both a mean index for respondents who answered at least four of the seven items and dichotomous (did experience any micro aggressions vs. no micro aggressions at all) variable.

**Sexual harassment and abuse of power.** Two single self-developed items on a 4-point scale (never—often) assessed whether participants had ever experienced anything they would consider a case of abuse of power or sexual harassment ("Have you experienced anything in the context of your artistic training, in the resulting practical activities or professional networks. . .? a). . .that you would describe as an abuse of power? b). . .that you would describe as sexual harassment or sexualized violence?"). We chose the broad framing in these two items deliberately to account for and capture a wide array of individual experiences. Conceptually, this aligns with the German General Equal Treatment Act, which explicitly provides victims with the interpretative sovereignty for the opening of sexual harassment complaint procedures [47].

**Sexual aggression.** We assessed sexual aggression experiences using the adapted Sexual Aggression and Victimization Scale (SAV-S) [11]. Four sexual aggression experiences (sexual touch, attempted intercourse, completed intercourse and "other", e.g. oral sex) were assessed separately for three different victim-perpetrator constellations (friend or acquaintance, current or former partners, strangers), resulting in overall 12 items referring to (lifetime) study context (e.g. "Has a friend or acquaintance within the artistic training context ever made you (or tried to make you) engage in sexual touching (kissing/petting) against your will?"). We informed that we were not aiming at experiences before the age of 14 years, as those would legally fall into the category of child sexual abuse. All experiences were assessed in a dichotomous response format (0 = no, 1 = yes). For statistical analyses, we calculated a dichotomous variable (did experience any sexual aggression vs. no sexual aggression at all).

**Technology-facilitated sexual violence.** These experiences were assessed using the adapted Technology-Facilitated Sexual Violence Scale, TFSV-V [10]. 19 items covering four technology-facilitated sexual violence domains (digital sexual harassment, image-based sexual abuse, sexual aggression/coercion, gender/sexuality-based harassment) from the original scale were used, e.g. "Someone has taken a nude or semi-nude picture of you within the artistic training context without your consent." Three original items referring to online gaming environments were not used for this study context. All experiences were assessed with dichotomous response options (0 = no, 1 = yes). Again, persons mentioning at least one experience within their educational context were presented follow-up questions [48, adapted]. For

statistical analyses, we calculated a dichotomous variable (did experience any technology-facilitated sexual violence vs. no technology-facilitated sexual violence at all).

**Support structure awareness.**   We assessed available awareness of supporting structures by asking whether participants knew of support structures to turn to when experiencing hostile behavior within their study context, in practical activities or professional networks. We offered a dichotomous response option (0 = no, 1 = yes). After choosing "yes", participants were invited to provide examples for support structures they would seek out in an open text field.

**Mental health, well-being, professional thriving, closeness with creative industries.**   To assess participants' mental health state, we used items from the 4-item Patient Health Questionnaire for Depression and Anxiety [49], e.g. "Over the past two weeks, how often have you been bothered by the following problems: Feeling nervous, anxious or on edge", 4-point rating "not at all"–"almost every day"). Additionally, we included items from the 5-item World Health Organization Well-Being Index [50], e.g. "Over the past two weeks, I have felt cheerful and in good spirits", 6-point rating "all of the time–at no time") to account for subjective well-being. Mean indices were calculated for participants who answered at least two of the four items.

To explore the impact of hostile behaviors on professional development and careers of early career artists, we assessed participants' sense of thriving in their respective educational context using the 10-item Thriving at Work Scale [51]; e.g. "In my subject and related practical activities, I find myself learning often", 5-point rating: don't agree at all–completely agree"). Mean indices were calculated for a minimum of 50% item response. We further used the Inclusion of Other in the Self scale (IOS) [52], a one-item pictorial task, to measure feeling of closeness with the creative industries. The IOS aims to depict respondents' closeness with an entity of interest by choosing one of seven combinations of incrementally overlapping circles (1 = no overlap, 7 = most overlap). We adapted both measures by specifically framing the item text with regard to the context of participants' study subject environments and creative industry context.

## Statistical analyses

All analyses were performed using IBM SPSS Statistics 29.0 and the PROCESS Macro [53].

## Results

The final convenience sample comprised $N = 611$ participants with a mean age of 27.5 years ($SD$ = 6.8). Participants reported being currently enrolled in a range of subject areas, resulting in the overarching clusters of music and musicology (36.0%), design (21.1%), fine arts (19.3%), performing arts, film, TV and theater studies (8.0%) and general arts (4.9%). The subject clusters presented here have been derived from the German Federal Statistics Office Student Register [54]. Further details on the specific fields of study included in each cluster can be found in the supporting information.

71.9% of the participants self-identified as female, trans, inter, non-binary or questioning their gender identity; 26.6% identified as male. 36.2% identified as lesbian, gay, bisexual, asexual or pansexual, making this an above-average diverse sample in terms of the sexual identities represented. 42.0% reported having diagnosed depression, anxiety or other mental health issues, while 31.8% reported having a chronic illness or longstanding health problem. Further socio-demographic details of the sample are presented in the supporting information.

### Experiences of hostile behaviors

Overall, almost all participants (91.8%, $N$ = 535) reported experiences of at least one form of hostile behavior within the academic context, ranging from more subtle forms such as micro

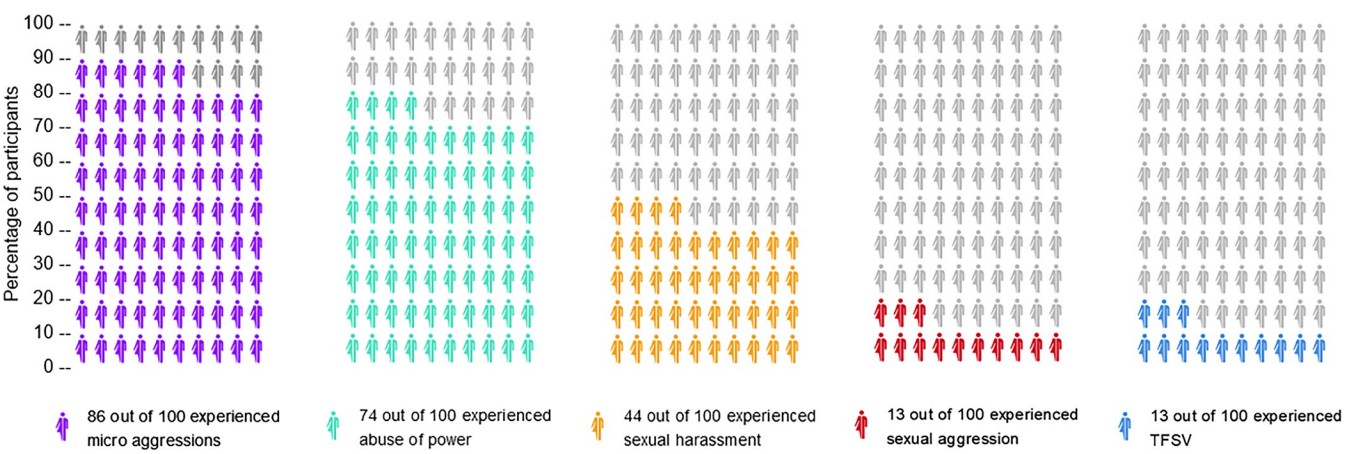

**Fig 2. Share of participants who experienced different forms of hostile behavior within the creative higher education context (in %, _N_ = 535).** TFSV: Technology-facilitated sexual violence.

aggressions to severe forms of sexual violence and harassment, both online and offline. Most commonly, participants reported micro aggressions (85.7%, _N_ = 491) and abuse of power (74.0%, _N_ = 527) (see Fig 2). Nearly half of all participants reported sexual harassment (43.9%, _N_ = 529). Nearly one eight reported sexual aggression (13.1%, _N_ = 467) and a similar rate reported technology-facilitated sexual violence (12.5%, _N_ = 455).

## Experiences by diversity domains

First, we explored our preregistered hypotheses on group differences in hostile behavior experiences. We run t-tests for independent samples for hostile experiences measured on continuous scales (micro aggressions, sexual harassment and abuse of power) and chi-square tests for hostile experiences measured in a dichotomous response format (sexual aggression and technology-facilitated sexual violence). We performed Spearman correlation analysis to investigate the association of participants' age and hostile behavior experience. A p-value of <0.05 (two-tailed) was considered significant. As secondary outcomes were considered conceptually explorative in nature, no adjustment for multiple testing was made.

As predicted, the observed pattern confirmed that marginalized participants reported experiences of hostile behavior more often than their colleagues from non-marginalized groups (see Table 1).

This difference was specifically pronounced for experiences of micro aggressions, abuse of power and sexual harassment. We found a similar pattern of results for more severe forms (sexual aggression and technology-facilitated sexual violence) of hostile behaviors, but the group differences were not statistically significant (apart from non-cis male persons being significantly more strongly affected by sexual aggression). Contrary to our predictions, participants' age was significantly positively correlated with reported experiences of micro aggressions ($r(491) = .14$, $p < .01$), abuse of power ($r(527) = .17$, $p < .001$) sexual harassment ($r(529) = .19$, $p < .001$), and sexual aggression ($r(467) = .09$, $p < .05$), while age was not correlated with experiences of technology-facilitated sexual violence.

## Experiences associated with mental health, professional thriving, and creative industry closeness

Second, we analyzed whether experiences of hostile behavior were linked to depressive symptoms, lower well-being, as well as lower professional thriving (see Table 2). As predicted, most

**Table 1. Experiences of hostile behavior by self-defined diversity domain membership.**

| | Micro aggressions (affectedness, 7 items, range 0–4) | | | Abuse of power (frequency, 1 item, range 1–4) | | | Sexual harassment (frequency, 1 item, range 1–4) | | | Sexual aggression (12 items, yes/no) | | | Technology-facilitated sexual violence (19 items, yes/no) | | |
|---|---|---|---|---|---|---|---|---|---|---|---|---|---|---|---|
| | t-tests of independence | | | | | | | | | $\chi^2$ tests of independence | | | | | |
| Domain | N | M(SD) | d | N | M(SD) | d | N | M(SD) | d | N | ≥1 yes n (%) | | N | ≥1 yes n (%) | |
| **Gender** | | | | | | | | | | | | | | | |
| FTINQS[1] | 361 | 1.31 (1.03) | 0.27** | 385 | 2.41 (1.04) | 0.21* | 386 | 1.70 (.89) | 0.28** | 461 | 53 (15.5) | $\chi^2 (1) =$ 7.21** | 450 | 46 (13.7) | $\chi^2 (1) =$ 1.34 |
| male | 125 | 1.03 (.97) | | 135 | 2.19 (.98) | | 136 | 1.46 (.69) | | | 7 (5.9) | | | 11 (9.6) | |
| **Sexual identity** | | | | | | | | | | | | | | | |
| LGBAPS | 182 | 1.31 (1.04) | 0.13 | 196 | 2.54 (1.02) | 0.29*** | 196 | 1.74 (.93) | 0.15 | 451 | 23 (13.2) | $\chi^2 (1) = 0.002$ | 439 | 28 (16.3) | $\chi^2 (1) =$ 3.63 |
| hetero | 288 | 1.18 (1.00) | | 308 | 2.25 (1.02) | | 309 | 1.61 (.81) | | | 37 (13.4) | | | 27 (10.1) | |
| **Care** | | | | | | | | | | | | | | | |
| yes | 44 | 1.59 (1.21) | 0.39* | 53 | 2.55 (1.12) | 0.23 | 52 | 1.92 (1.00) | 0.35* | 459 | 6 (13.6) | $\chi^2 (1) = 0.005$ | 447 | 6 (14.0) | $\chi^2 (1) =$ 0.09 |
| no | 438 | 1.20 (1.00) | | 466 | 2.32 (1.01) | | 468 | 1.62 (.83) | | | 55 (13.3) | | | 50 (12.4) | |
| **Migration history** | | | | | | | | | | | | | | | |
| yes | 136 | 1.35 (1.05) | 0.15 | 146 | 2.27 (1.02) | -0.10 | 146 | 1.62 (.82) | -0.05 | 466 | 20 (16.0) | $\chi^2 (1) = 1.27$ | 454 | 12 (10.2) | $\chi^2 (1) =$ 0.83 |
| no | 354 | 1.20 (1.01) | | 379 | 2.37 (1.02) | | 381 | 1.66 (.86) | | | 41 (12.0) | | | 45 (13.4) | |
| **Ethnic-racial identity** | | | | | | | | | | | | | | | |
| marginalized | 102 | 1.52 (1.08) | 0.38*** | 113 | 2.49(1.05) | 0.12 | 112 | 1.76 (.86) | 0.13 | 379 | 17 (17.7) | $\chi^2 (1) = 2.96$ | 371 | 10 (11.0) | $\chi^2 (1) =$ 0.41 |
| non-marginalized | 298 | 1.14 (.98) | | 313 | 2.37(1.01) | | 315 | 1.65 (.86) | | | 31 (11.0) | | | 38 (13.6) | |
| **Mental health issues** | | | | | | | | | | | | | | | |
| yes | 212 | 1.40 (1.05) | 0.29*** | 229 | 2.48 (1.04) | 0.25** | 231 | 1.79 (.90) | 0.31*** | 461 | 34 (16.5) | $\chi^2 (1) = 3.47$ | 449 | 30 (15.1) | $\chi^2 (1) =$ 2.66 |
| no | 272 | 1.11 (.98) | | 290 | 2.23 (1.00) | | 290 | 1.53 (.80) | | | 27 (10.6) | | | 25 (10.0) | |
| **Disability** | | | | | | | | | | | | | | | |
| yes | 22 | 1.68 (1.18) | 0.46* | 26 | 2.50 (1.07) | 0.16 | 26 | 2.08 (1.26) | 0.54 | 459 | 5 (25.0) | $\chi^2 (1) = 2.62$ | 447 | 3 (15.8) | $\chi^2 (1) =$ 0.17 |
| no | 460 | 1.21 (1.01) | | 490 | 2.33 (1.03) | | 493 | 1.62 (.82) | | | 55 (12.5) | | | 54 (12.6) | |
| **Physical health issues** | | | | | | | | | | | | | | | |
| yes | 161 | 1.37 (1.02) | 0.21* | 175 | 2.49 (1.00) | 0.21* | 175 | 1.69 (.88) | 0.06 | 457 | 21 (14.0) | $\chi^2 (1) = 0.24$ | 447 | 18 (12.2) | $\chi^2 (1) =$ 0.03 |
| no | 321 | 1.16 (1.01) | | 340 | 2.28 (1.03) | | 342 | 1.64 (.84) | | | 38 (12.4) | | | 38 (12.7) | |

Note.

*$p < .05$

** $p < .01$

***$p < .001$.

[1]FTINQS: Female, trans, inter, non-binary, queer, self-identified; LGBAPS: Lesbian, gay, bisexual, asexual, pansexual, self-identified.

**Table 2. Associations of experiences of hostile behavior with mental health, professional thriving and closeness with the creative industries.**

| | N | M | SD | Micro aggression | Abuse of power | Sexual harassment | Sexual aggression | Technology-facilitated sexual violence |
|---|---|---|---|---|---|---|---|---|
| **Depressive symptoms** | 539 | 2.15 | 0.75 | .29*** | .16*** | .12** | .08 | .11* |
| **Low well-being** | 534 | 3.58 | 1.05 | .30*** | .20*** | .12** | .12** | .08 |
| **Professional thriving** | 442 | 3.81 | 0.63 | -.37*** | -.24*** | -.16*** | -.13** | -.16** |
| **Closeness with creative industries** | 447 | 4.36 | 1.56 | -.06 | -.01 | .02 | .05 | .08 |

*Note.* Correlations' significance levels

*$p < .05$

** $p < .01$

***$p < .001$.

hostile behavior experiences correlated significantly with higher depressive symptoms, lower well-being and lower professional thriving. There was no association between experiences of sexual aggression and depressive symptoms. Similarly, there was no association between experiences of technology-facilitated sexual violence and lower well-being. We also found no significant associations between experiences of hostile behavior and participants' reported closeness with the creative industries.

## Moderated associations

Finally, we examined if diversity domain membership moderates the link between hostile behavior experiences and mental health and professional thriving. We did not examine moderating links with closeness with the creative industries, as no significant correlations had been found in the first place. We ran a series of regression analyses (enter method) with the respective mental health or thriving aspect as an outcome, predicted by each diversity domain (being a member of a privileged vs. marginalized group). We also included the interaction term of each diversity domain with the respective hostile behavior (group mean centered continuous variables) as predictors. Contrary to our predictions, these analyses revealed mixed and mostly non-significant results. No clear moderation pattern emerged. All results are reported in the supporting information.

## Exploratory analyses

In addition to the pre-registered hypotheses, we conducted some exploratory analyses.

### Experiences of discrimination and awareness of support structures

Participants were asked to list discrimination experiences associated with self-defined diversity domain membership. Many participants reported experiencing at least one form of discrimination (55.8% of 564 participants) during the last 24 months, and still a sizeable proportion several forms of discrimination (23.7%). The most commonly reported discrimination experiences were due to gender (26.8%), low income (9.8%), and racist attitudes of others (9.2%). The most common combination of discriminatory experiences was due to gender and low income. 52 participants chose to enter individual descriptions of discrimination experiences that were not captured by the list (6.6%); among them body appearance or looks (13.5%) were often mentioned (see supporting information).

We further analyzed whether participants were aware of counseling services that they could turn to in cases of hostile behavior incidents. Of 531 participants, 42.7% stated they didn't know of any structures within their academic education context that offered such support.

## Mediating role of experiences of hostile behaviors

We explored whether experiences of hostile behaviors might mediate the association between belonging to a marginalized group and worse mental health and professional development aspects. To this end, we first analyzed whether there were group differences in mental health and professional development aspects (see Table 3). The pattern of results was mixed. Identification with a marginalized group (e.g. as non-male, non-heterosexual, and having mental and physical health issues) showed associations with some, but not all outcomes (in particular depressive symptoms and well-being).

Next, we conducted bootstrapped (5000 iterations) mediation analyses with PROCESS 4.2 (model 4) to investigate experiences of hostile behavior as possible mediators of marginalized groups' worse mental health and worse professional development aspects. Since the number of models was very high, we exemplary report selected results in the next paragraphs. We provide the results of all remaining mediation analyses in the supporting information.

Most prominently, we found micro aggressions experience to explain substantial parts of worse mental health and thriving outcomes of early career artists from various marginalized groups: gender and ethnic-racial minorities, as well as persons marginalized in terms of mental and physical health diagnoses and disabilities. Fig 3 provides an example. The mediation model examining the association between being of a gender identity other than cis male and reporting depressive symptoms, mediated by micro aggressions, revealed a significant indirect effect of $b = -0.06$, $SE = 0.02$, 95% CI = $[-0.11, -0.02]$, explaining a third of the direct significant effect of gender marginalization on depressive symptoms ($b = -0.20$, $SE = 0.07$, 95% CI = $[-0.34; -0.05]$).

In similar models, significantly lower levels of well-being of non-cis male participants ($b = -0.22$, $SE = 0.10$, 95% CI = $[-0.42; -0.15]$, $p < .05$) were explained by experiencing abuse of power to almost a fifth: ($b = -0.04$, $SE = 0.02$, 95% CI = $[-0.09; -0.004]$). Lower well-being of non-heterosexual participants ($b = -0.23$, $SE = 0.09$, 95% CI = $[-0.41; -0.04]$, $p < .05$) was explained by experiencing abuse of power to almost a quarter: ($b = -0.06$, $SE = 0.02$, 95% CI = $[-0.10; -0.018]$). For professional thriving levels of non-heterosexual participants, mediation analysis yielded an indirect effect of abuse of power experience ($b = 0.03$, $SE = 0.02$, 95% CI = $[0.01; 0.07]$), while the direct effect of sexual identity was nonsignificant ($b = 0.08$, $SE = 0.06$,

**Table 3. Associations of diversity domains with mental health, professional thriving and closeness with the creative industries.**

|  | N | M | SD | Gender Identity | Sexual Identity | Age | Care | Migration history | Ethnic-racial identity | Mental health | Disability | Physical health |
|---|---|---|---|---|---|---|---|---|---|---|---|---|
| **Depressive symptoms** | 539 | 2.15 | .75 | -.14** | -.13** | -.09* | -.02 | -.06 | -.10* | -.27*** | -.07 | -.11* |
| **Low well-being** | 534 | 3.58 | 1.05 | -.11* | -.13** | .02 | -.04 | -.01 | -.06 | -.25*** | -.11* | -.17*** |
| **Professional thriving** | 442 | 3.81 | .63 | .08 | .09 | .02 | .07 | .06 | .06 | .24*** | .19*** | .19*** |
| **Closeness with creative industries** | 447 | 4.36 | 1.56 | .05 | .05 | -.02 | .05 | -.07 | -.02 | .12* | .05 | .08 |

*Note*. For all diversity domains, marginalized group (e.g. gender: female, inter, trans, non-binary, questioning) coded as 0 and non-marginalized group (e.g. gender: male) coded as 1.

*$p < .05$

** $p < .01$

***$p < .001$.

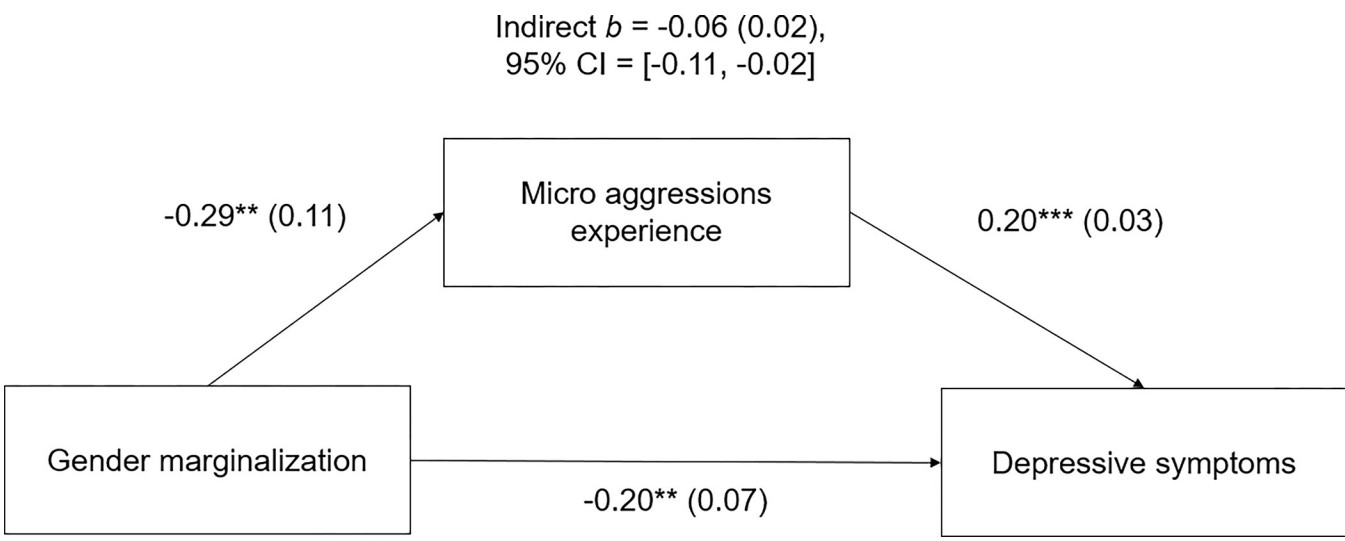

**Fig 3. Mediation model showing effect of gender marginalization on depressive symptoms as mediated by micro aggressions experience.** Indirect effects shown above the model. Unstandardized coefficients given outside parentheses, standard errors in parentheses. Asterisks indicate levels of significance (**$p <$ .01*, *** $p < .001$). CI = confidence interval.

95% CI = [-0.04; 0.21]). An indirect effect of sexual harassment experience ($b = 0.03$, $SE = 0.01$, 95% CI = [0.01; 0.06]) explained lower levels of professional thriving of non-cis male persons to almost a third (nonsignificant direct effect of gender: $b = 0.09$, $SE = 0.07$, 95% CI = [-.05; 0.23]).

## Discussion

Our study addresses the research gap regarding early career artists' experiences of hostile behaviors and quantifies such experiences endured by students at German art and music HEIs. Our findings demonstrate that students belonging to marginalized groups are more likely to report having such experiences, and that they correlate with decreased mental health and well-being. An alarming percentage of violent and discriminatory experiences were reported within creative HEI contexts: the majority of participants reported experiences of micro aggressions, almost every second person reported sexual harassment, and one of eight respondents reported more severe forms of sexual aggression. In addition, 40 percent of the respondents did not have sufficient knowledge of support structures.

While we do not draw causal conclusions from our findings, our results indicate that early career artists from marginalized groups report experiencing hostile behaviors more frequently than non-marginalized persons. Individuals not identifying as cis gender men and persons with mental health issues appeared to be particularly prone to such experiences. In addition, experiences of micro aggressions and abuse of power were widespread and significantly associated with reduced mental health and professional thriving in this sample. The mediating role of such experiences, as revealed in our exploratory analyses, highlights the risk for adverse impacts on professional thriving and mental health outcomes for affected persons, particularly those who are less privileged and thus more vulnerable to experiences of hostile behavior in the first place.

Our findings suggest higher rates of sexual harassment (44% vs. 27%) and interpersonal boundary violations (86% micro aggressions vs. 42% bullying and 57% inappropriate behavior) than a similar prior study from the UK [38]. However, a recent study on gender-based violence in European HEIs [55] also found rather high rates of violent experiences. 62% of their

participants reported one or more violent experiences, with members of minority groups reporting higher rates of victimization. In addition, our results are consistent with previous evidence for low utilization of complaint structures within academic institutions [56, 57]. They also are also in line with findings suggesting that multi-layered discrimination affects creative workers at all career stages [32, 33].

## Strengths and limitations

Our study has several strengths. First, its focus on hostile behaviors in combination with the marginalization and mental health of early career artists means it fills an existing gap particularly within the empirical literature on creative higher education. A better understanding of the conditions and factors influencing early career artists' mental health is a societal responsibility and an important step towards improving working conditions in the present and future creative sector. Second, our study considers a wide range of hostile behaviors, drawing on the concept of violence as a continuum [7]. Following this approach, hostile institutional climates in specific areas can be identified at early stages. Thus, the normalization of such behaviors in creative higher education is countered in a timely manner, preventing more severe violence and assault [15]. This, in turn, can contribute to a better understanding and future improvement of work cultures in the cultural and creative industries. Third, this research follows a diversity-focused approach, centering on the experiences of marginalized groups. This is especially valuable for the creative field, which claims to be at the forefront of progressive views on societal challenges, even though it maintains access barriers, career disadvantages, and discrimination against marginalized individuals [33, 42]. By assessing various diversity domains, we provide a comprehensive insight into the experiences of marginalized creatives at early career stages, alongside their mental health status.

While our study focused on the vulnerabilities of marginalized groups to hostile behavior, we do not claim causality in the observed patterns due the cross-sectional, self-report design. It is possible that individuals who were sensitive to diversity and discrimination were particularly motivated to participate or prone to interpret negative experiences as resulting from marginalization. We also cannot exclude the possibility that third variables drive the pattern we observed in our data. However, we sought to appeal to the whole student body regardless of individual experiences by inviting them to participate in the study in a neutral manner while responsibly disclosing potentially distressing content, and offering an incentive in the form of a prize draw.

To our surprise and contrary to our assumptions, we did not find any significant associations between the experience of hostile behavior and reported feelings of closeness with the creative industries. There are several possible explanations for this. On the one hand, this result may be methodologically based on the selection of the measurement instrument, which was developed for the investigation of feelings of closeness between individuals and groups [52]. It is possible that the use is not specific enough if the "other" surveyed represents the creative industries community. Future research could address this question more specifically in order to optimize the assessment of the professional affiliation as an artist.

On the other hand, the generally strong ideational identification with the creative industries [25, 27] mentioned earlier may mean that people do not associate the experience of hostile behavior with the fundamental desire to work in this industry. Professional narratives of enduring boundary violations as an inherent part of being an artist and tolerating problematic situations [23] could potentially serve as a buffer against distancing from this working environment. An in-depth mixed-method study could be of interest here.

Apart from quantifying experiences of hostile behavior, future studies should further examine victim-perpetrator constellations, and situational characteristics [11], to address how and where specific positions of power come into play [32]. While teaching practices, inherent physical proximity, and field-specific challenges can differ greatly in art and music education, we did not differentiate between artistic subject groups for data protection reasons. Future research might consider this, and focus on intersections of diversity domains, for which the current sample was too small.

## Implications for interventions in creative higher education

Our study is an initial needs assessment that quantifies reported experiences of hostile behaviors at art and music HEIs. However, further research needs to go beyond describing the extent of the problem and delve more deeply into the structural power mechanisms behind specific experiences reported here. Hostile behaviors in the creative academic sphere need to be explored more closely and discussed across intersecting axes of power and privilege, including gender, race, health status, and sexual identity. Given the need for evidence-based prevention at art and music HEIs [37], prevalent hostile behaviors should be examined more indepth with regards to specific situational risk assessment protocols and possible leverage points. One possible approach is to target collective scripts and norms [12] related to these behaviors, as well as overall concepts of power, appropriate behavior and interpersonal boundaries in creative professions [23, 25]. Possible starting points include raising broad awareness of what exactly constitutes harassment and professional misconduct within creative higher education structures, as well as providing appropriate formats and time for reflection on power dynamics for both students and staff [34]. Finally, specific sanctions for hostile behaviors and the responsible authorities for prosecution must be defined in a transparent and binding manner. All members of creative HEIs must be informed about their rights and obligations.

Interventions should be monitored longitudinally, and repeated surveys of a broad range of hostile behaviors should be conducted at the institutional level. Our findings suggest that interventions in this already highly competitive professional environment need to be designed with particular sensitivity to groups marginalized due to gender identity and mental health issues to reduce minority stress and social unsafety [19, 20].

According to our research, early career artists report high rates of hostile behavior experiences and limited knowledge of complaint structures. A lack of awareness of support services and students' rights and a reluctance to report may stem from a lack of trust in institutions [56]. This can be accompanied by fear of exacerbating one's situation and the expectation that those affected will not be believed [57]. Arts HEIs might thus benefit from external counseling services due to their unique institutional environments with close relationships and hierarchical structures [34]. We summarize our suggested interventions for creative HEIs in Table 4.

## Conclusion

Our research suggests that experiences of hostile behavior permeate the professional environments and early career stages of young and marginalized artists, beginning much earlier than their first positions as trained professionals in the creative sector. To prevent the problems addressed by the #MeToo movement from starting so early, and to avoid potentially damaging the mental health and personal development of young artists, the culture and creative industries need specifically sector-tailored prevention and intervention strategies that target entire organizations at all levels and age groups. This is particularly urgent for art and music schools, which set the scene and play a critical role in shaping future careers and networks of aspiring artists. As our data demonstrates, it is of utmost importance to consider intersecting and

**Table 4. Suggested interventions in the field of creative higher education.**

| Field | Suggested interventions |
|---|---|
| **Preventative measures** | Situational risk assessment |
| | Measures targeting normative professional beliefs |
| | Specification of relevant terms and regulations regarding professional conduct |
| | Reflective power-critical formats for different target audiences |
| | Clear and binding formulation of sanctions |
| | Repeated and longitudinal monitoring of measures |
| | Tailored interventions for marginalized groups |
| **Support service utilization** | Raising awareness regarding internal support services |
| | Raising awareness regarding students' rights |
| | Transparent information about complaint processes |
| | Independent, external counseling services |

multiple forms of marginalization and individual positions within contexts of power when designing supportive and preventive strategies. It is also crucial to account for the different effects that hostility and violence within the academic context can have on the mental health and opportunities to thrive experienced by art and music students who already experience different forms of marginalization. Future sector-oriented and institutional policies need to sustainably ensure safer creative contexts from the outset, especially for marginalized artists.

## Supporting information

**S1 Table. Study subject clusters within creative higher education according to German federal office of statistics.**
(DOCX)

**S2 Table. Sociodemographic sample characteristics.**
(DOCX)

**S3 Table. Experiences of discrimination.**
(DOCX)

**S4 Table. Moderating effects of diversity domains on association of hostile behavior experience with mental health and professional thriving.**
(DOCX)

**S5 Table. Micro aggressions experience mediating association of diversity domains with mental health, thriving and industry closeness.**
(DOCX)

**S6 Table. Abuse of power experience mediating association of diversity domains with mental health, thriving and industry closeness.**
(DOCX)

**S7 Table. Sexual harassment experience mediating association of diversity domains with mental health, thriving and industry closeness.**
(DOCX)

**S8 Table. Sexual aggression experience mediating association of diversity domains with mental health, thriving and industry closeness.**
(DOCX)

**S9 Table. Technology-facilitated sexual violence experience mediating association of diversity domains with mental health, thriving and industry closeness.**
(DOCX)

## Acknowledgments

First and foremost, our heartfelt thanks to all the participants who contributed to this research by sharing their experiences, and to the German art and music institutions and networks that supported the study by disseminating it. Many thanks to the team of the Gender in Medicine Lab at Charité–Universitätsmedizin Berlin, and the team of the *Good Work in a Transformative World* program at the Berlin Social Science Center for their insightful feedback on the manuscript. We are also grateful to the Berlin Institute of Health–Medical Informatics Team for advice on data de-identification.

## Author Contributions

**Conceptualization:** Marina Fischer, Susanne Veit, Gertraud Stadler.

**Data curation:** Marina Fischer.

**Formal analysis:** Marina Fischer, Gertraud Stadler.

**Funding acquisition:** Marina Fischer, Susanne Veit.

**Investigation:** Marina Fischer, Gertraud Stadler.

**Methodology:** Marina Fischer, Susanne Veit.

**Project administration:** Marina Fischer, Pichit Buspavanich.

**Resources:** Marina Fischer, Gertraud Stadler.

**Supervision:** Susanne Veit, Pichit Buspavanich, Gertraud Stadler.

**Visualization:** Marina Fischer, Gertraud Stadler.

**Writing – original draft:** Marina Fischer.

**Writing – review & editing:** Marina Fischer, Susanne Veit, Pichit Buspavanich, Gertraud Stadler.

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
