## [Decision Letter · Decision Letter 0]

8 Oct 2024

PONE-D-24-25065Turning the spotlight: Hostile behavior in creative higher education and links to mental health in marginalized groups.PLOS ONE

Dear Dr. Fischer,

Thank you for submitting your manuscript to PLOS ONE. After careful consideration, we feel that it has merit but does not fully meet PLOS ONE’s publication criteria as it currently stands. Therefore, we invite you to submit a revised version of the manuscript that addresses the points raised during the review process.

We look forward to receiving your revised manuscript.

Kind regards,

Vincenzo Auriemma

Academic Editor

PLOS ONE

Journal Requirements:

Additional Editor Comments :

Dear Author, based on the revisions received, I suggest you read and check if the minor revisions requested can be fulfilled. In this way the work can receive greater appeal.

Reviewers' comments:

Reviewer's Responses to Questions

**Comments to the Author**

1. Is the manuscript technically sound, and do the data support the conclusions?

Reviewer #1: Yes

Reviewer #2: Yes

2. Has the statistical analysis been performed appropriately and rigorously? 

Reviewer #1: Yes

Reviewer #2: Yes

3. Have the authors made all data underlying the findings in their manuscript fully available?

Reviewer #1: Yes

Reviewer #2: Yes

4. Is the manuscript presented in an intelligible fashion and written in standard English?

Reviewer #1: Yes

Reviewer #2: Yes

5. Review Comments to the Author

Reviewer #1: This is an insightful and important paper with a rich data base, clear figures, and overall, well written. Some suggested edits for structure/format are below:

In the "Perspectives on hostile behaviors" section - opening 2 sentences are not a complete paragraph. I suggest moving the last paragraph in the section (currently bottom of p.4) to the end of the second opening sentence, thus making a full paragraph and improving the flow of the first main section.

I was surprised no significant association between hostile behavior and participants' reported closeness with the creative industries - this could be unpacked more in the discussion and/or on p. 18.

Very long sub-heading on p. 19 serves more like a topic sentence than a sub-heading. I encourage the authors to revisit their subheadings - there are many of them, sometimes with sections only 1 paragraph long, and some subheadings, like the one on p.19, are serving the purpose of a topic sentence rather than a sub-heading. Similarly, the "Support Structure Awareness" section on p. 20 is only 2 sentences long. This should be revised.

Discussion section is strong and use of important concepts/terms such as a continuum view of violence and use of diversity domains are a strength of the paper.

Interventions listed on pp. 25-26 are important. I suggest the sub-heading is revised to highlight this is a section focused on recommended interventions. I also suggest a table or bulleted list to highlight the important takeaway suggestions for readers who are in a position to implement such interventions to clearly be able to see and remember them. Currently they could easily be lost during a quick read.

Reviewer #2: The article is particularly interesting. However, in several places in it there are portions of text that could be merged to make the text uniform. Likewise, the titles would need to be reworded to make them more appealing. Finally, I recommend a re-reading of the entire text to correct the remaining typos.

6. PLOS authors have the option to publish the peer review history of their article (what does this mean?). If published, this will include your full peer review and any attached files.

Reviewer #1: No

Reviewer #2: **Yes: **Luisa Nardi

---

## [Author Response · Author response to Decision Letter 0]

19 Nov 2024

Dear Vincenzo Auriemma,

Thank you very much for giving us the opportunity to revise our manuscript Turning the spotlight: Hostile behavior in creative higher education and links to mental health in marginalized groups by Marina Fischer, Susanne Veit, Pichit Buspavanich, and Gertraud Stadler.

We highly appreciate your interest in our work and are grateful to the reviewers for their insightful and constructive comments. In light of the constructive feedback, we have thoroughly revised our manuscript, taking into account the suggestions put forth by Reviewer 1 and Reviewer 2. Notes regarding each point raised are included in this document. 

We uploaded both a marked-up copy with highlighted changes to the manuscript, and an unmarked version to the PLOS One Editorial Manager. Please note that owing to partial restructuring and moving of text passages, some references in the reference list have been renumbered. These changes can also be tracked in highlighted mode. There have been no other changes to the reference list. 

In order to comply with open research principles while adhering to the data protection restrictions set forth by the Ethics Board of our institution (Charité - Universitätsmedizin Berlin, https://ethikkommission.charite.de/en/), we provide a de-identified data slice containing relevant variables that have been thoroughly anonymized. We submit a comprehensive data protection protocol in conjunction with our manuscript. The data slice alongside study materials is accessible via the OSF repository: https://osf.io/stced.

As some of the reviewers’ comments concerned issues of language and style, thorough professional language editing has been applied throughout the whole manuscript in order to enhance the appeal even more. 

We value your further consideration of our manuscript for publication in PLOS One.

Please do not hesitate to contact us with any questions.

Sincerely,

Marina Fischer

In the following section, we present a point-by-point response to the reviewer's comments. We would like to express our gratitude to Reviewer 1 and Reviewer 2 for dedicating their valuable time and providing insightful contributions. We are most grateful for the helpful feedback you have provided, which has undoubtedly enhanced the quality of our manuscript. Please find below point-by-point responses to the comments that have been provided. All amendments are indicated in the marked-up version of the revised manuscript.

Reviewer #1: 

This is an insightful and important paper with a rich data base, clear figures, and overall, well written. 

Response: We are grateful for the appreciative feedback from Reviewer 1 regarding our work.

Some suggested edits for structure/format are below: In the "Perspectives on hostile behaviors" section - opening 2 sentences are not a complete paragraph. I suggest moving the last paragraph in the section (currently bottom of p.4) to the end of the second opening sentence, thus making a full paragraph and improving the flow of the first main section.

Response: We thank Reviewer 1 for the helpful comment on the text structure, which we have changed accordingly. The paragraph can now be found on p.3.

I was surprised no significant association between hostile behavior and participants' reported closeness with the creative industries - this could be unpacked more in the discussion and/or on p. 18.

Response: We fully agree with this comment and are grateful for the detailed interest in our research results. We also believe that this aspect should receive more attention. We have therefore gladly taken up the suggestion and discussed some possible interpretations of this finding on p. 25-26.

Very long sub-heading on p. 19 serves more like a topic sentence than a sub-heading. I encourage the authors to revisit their subheadings - there are many of them, sometimes with sections only 1 paragraph long, and some subheadings, like the one on p.19, are serving the purpose of a topic sentence rather than a sub-heading. 

Response: We would like to thank Reviewer 1 for the important reference to patterns in our manuscript composition and the helpful suggestions for improvement. In the light of this valuable advice, we have amended the subheading on p. 19. In addition, we have checked all subheadings of the manuscript for comprehensibility and necessity as well as embedding in the overall text. Accordingly, we have adapted the headings on p. 3, p. 5, p. 6, pp. 11-12, p. 15, pp. 18-21, and p. 26, or revised them for better comprehensibility. We also used the consultation of professional language editing services for this purpose.

Similarly, the "Support Structure Awareness" section on p. 20 is only 2 sentences long. This should be revised.

Response: In response to this constructive observation, we consolidated two paragraphs from the "Exploratory analyses" section. The aforementioned subsection is now presented under the subheading "Experiences of discrimination and awareness of support structures" (p. 20).

Discussion section is strong and use of important concepts/terms such as a continuum view of violence and use of diversity domains are a strength of the paper.

Response:

We greatly appreciate the positive evaluation of Reviewer 1 regarding the discussion section.

Interventions listed on pp. 25-26 are important. I suggest the sub-heading is revised to highlight this is a section focused on recommended interventions. 

Response: We thank Reviewer 1 for underscoring the importance of the proposed interventions and have revised the subheading on p. 26 accordingly. 

I also suggest a table or bulleted list to highlight the important takeaway suggestions for readers who are in a position to implement such interventions to clearly be able to see and remember them. Currently they could easily be lost during a quick read.

Response: We agree with Reviewer 1's comment and thank you for pointing this out and proposing helpful solutions. In fact, the proposed interventions may not have received enough attention in the original discussion section. We have therefore decided to include Table 4 on p. 28, where we clearly present our intervention recommendations. Here we further differentiate the intervention approaches listed in the body text according to the categories “Preventative measures” and “Support service utilization” in order to further improve the overview.

Reviewer #2: 

The article is particularly interesting. However, in several places in it there are portions of text that could be merged to make the text uniform. 

Response: We would like to thank Reviewer 2 for the overall positive evaluation of our manuscript. 

To follow your suggestion to improve the structure of the text, we have used the services of a professional academic language editor. This has resulted in rewording and shortening of overly long and incomprehensible sentences or paragraphs throughout. We hope that this will improve the readability and consistency of the text.

Likewise, the titles would need to be reworded to make them more appealing. 

Response: We have revised (sub)titles in line with the comments of both reviewers: On p. 3, p. 5, p. 6, pp. 11-12, p. 15, pp. 18-21, and p. 26, we have either shortened titles and/or subjected them to a further critical reading and rewording in order to make them more appealing and better embedded in the narrative flow of the manuscript.

Finally, I recommend a re-reading of the entire text to correct the remaining typos.

Response: We would like to thank Reviewer 2 for their careful reading of our manuscript. We have thoroughly proofread the entire text for typos. The corrections can be found in the marked-up version of the resubmitted manuscript.

---

## [Editor Report · Decision Letter 1]

21 Nov 2024

Turning the spotlight: Hostile behavior in creative higher education and links to mental health in marginalized groups.

PONE-D-24-25065R1

Dear Dr. Fischer,

We’re pleased to inform you that your manuscript has been judged scientifically suitable for publication and will be formally accepted for publication once it meets all outstanding technical requirements.

Kind regards,

Vincenzo Auriemma

Academic Editor

PLOS ONE

---

## [Editor Report · Acceptance letter]

18 Dec 2024

PONE-D-24-25065R1 

PLOS ONE

Dear Dr. Fischer, 

I'm pleased to inform you that your manuscript has been deemed suitable for publication in PLOS ONE. Congratulations! Your manuscript is now being handed over to our production team.

Kind regards, 

on behalf of

Dr. Vincenzo Auriemma 

Academic Editor

PLOS ONE